# Regional and country-level trends in cervical cancer screening coverage in sub-Saharan Africa: A systematic analysis of population-based surveys (2000–2020)

Lily Yang[1], Marie-Claude Boily[2], Minttu M. Rönn[3], Dorcas Obiri-Yeboah[4], Imran Morhason-Bello[5], Nicolas Meda[6], Olga Lompo[7], Philippe Mayaud[8], Michael Pickles[2], Marc Brisson[9], Caroline Hodgins[10], Sinead Delany-Moretlwe[11], Mathieu Maheu-Giroux[1]*

1 Department of Epidemiology and Biostatistics, School of Population and Global Health, McGill University, Montréal, Québec, Canada, 2 Medical Research Council Centre for Global Infectious Disease Analysis, School of Public Health, Imperial College London, London, United Kingdom, 3 Department of Global Health and Population, Harvard T.H. Chan School of Public Health, Boston, Massachusetts, United States of America, 4 Microbiology and Immunology Department, School of Medical Sciences, College of Health and Allied Sciences, University of Cape Coast, Cape Coast, Ghana, 5 Department of Obstetrics and Gynaecology and Institute of Advanced Medical Research and Training, College of Medicine, University College Hospital, University of Ibadan, Ibadan, Nigeria, 6 Faculty of Medicine, Université Joseph Ki-Zerbo, Ouagadougou, Burkina Faso, 7 Centre de Recherche Internationale en Santé, Université de Ouagadougou, Ouagadougou, Burkina Faso, 8 Clinical Research Department, London School of Hygiene & Tropical Medicine, London, United Kingdom, 9 Centre de recherche du CHU de Québec-Université Laval, Québec, Québec, Canada; Département de médecine sociale et préventive, Faculté de médecine, Université Laval, Québec, QC, Canada, 10 Department of Microbiology and Immunology, McGill University, Montréal, Québec, Canada, 11 Wits Reproductive Health and HIV Institute, University of Witwatersrand, Johannesburg, South Africa

* mathieu.maheu-giroux@mcgill.ca

**Data Availability Statement:** The data underlying the results presented in the study can be freely requested, under some conditions, from The DHS

## Abstract

### Background

Sub-Saharan Africa (SSA) has the highest cervical cancer (CC) burden globally—worsened by its HIV epidemics. In 2020, the World Health Organization (WHO) introduced a CC elimination strategy with goals for vaccination, screening, and treatment. To benchmark progress, we examined temporal trends in screening coverage, percent screened at least twice by the age of 45, screening coverage among women living with HIV (WLHIV), and pre-cancer treatment coverage in SSA.

### Methods and findings

We conducted a systematic analysis of cross-sectional population-based surveys. It included 52 surveys from 28 countries (2000 to 2020) with information on CC screening among women aged 25 to 49 years ($N$ = 151,338 women). We estimated lifetime and past 3-year screening coverage by age, year, country, and HIV serostatus using a Bayesian multilevel model. Post-stratification and imputations were done to obtain aggregate national, regional, and SSA-level estimates. To measure re-screening by age 45, a life table model

Program (https://dhsprogram.com/), The PHIA Project (https://phia.icap.columbia.edu/), the WHO Multi-Country Studies Data Archive (https://apps.who.int/healthinfo/systems/surveydata/index.php/catalog), the WHO NCD microdata repository (https://extranet.who.int/ncdsmicrodata/index.php/home), the Kenya National Data Archive (KeNADA) (https://statistics.knbs.or.ke/nada/index.php/catalog), and the Human Sciences Research Council of South Africa (http://datacuration.hsrc.ac.za/). How to access and request sources of data for this analysis can also be found in S1 Table in the supporting information. A cleaned dataset alongside code used for the project can be found here: https://github.com/pop-health-mod/cc-screening.

**Funding:** This work was funded by the Canadian Institutes of Health Research (to MM-G, MCB, PM, MP, MB, SD-M - https://webapps.cihr-irsc.gc.ca/decisions/p/project_details.html?applId=401978&lang=en) and supported by a Canada Research Chair (Tier 2) in Population Health Modeling (to MM-G - https://webapps.cihr-irsc.gc.ca/decisions/p/project_details.html?applId=415376&lang=en). Support was also received through a Master's award from the Canada Graduate Scholarships funded by the Canadian Institutes of Health Research (to LY - https://www.nserc-crsng.gc.ca/students-etudiants/pg-cs/cgsm-bescm_eng.asp). Additionally MCB acknowledges funding from the Medical Research Council (MRC) Centre for Global Infectious Disease Analysis (MR/R015600/1 - https://www.ukri.org/councils/mrc/), jointly funded by the UK MRC and the UK Foreign, Commonwealth & Development Office (FCDO), under the MRC/FCDO Concordat agreement and is also part of the EDCTP2 programme supported by the European Union. The funders had no role in study design, data collection and analysis, decision to publish, or preparation of the manuscript.

**Competing interests:** The authors have declared that no competing interests exist.

**Abbreviations:** CC, cervical cancer; COVID-19, Coronavirus Disease 2019; CrI, credible interval; HPV, human papillomaviruses; VIA, visual inspection with acetic acid; WHO, World Health Organization; WLHIV, women living with HIV.

was developed. Finally, self-reported pre-cancer treatment coverage was pooled across surveys using a Bayesian meta-analysis. Overall, an estimated 14% (95% credible intervals [95% CrI]: 11% to 21%) of women aged 30 to 49 years had ever been screened for CC in 2020, with important regional and country-level differences. In Eastern and Western/Central Africa, regional screening coverages remained constant from 2000 to 2020 and WLHIV had greater odds of being screened compared to women without HIV. In Southern Africa, however, screening coverages increased and WLHIV had equal odds of screening. Notably this region was found to have higher screening coverage in comparison to other African regions. Rescreening rates were high among women who have already been screened; however, it was estimated that only 12% (95% CrI: 10% to 18%) of women had been screened twice or more by age 45 in 2020. Finally, treatment coverage among 4 countries with data was 84% (95% CrI: 70% to 95%). Limitations of our analyses include the paucity of data on screening modality and the few countries that had multiple surveys.

## Conclusion

Overall, CC screening coverage remains sub-optimal and did not improve much over the last 2 decades, outside of Southern Africa. Action is needed to increase screening coverage if CC elimination is to be achieved.

## Author summary

### Why was this study done?

- Cervical cancer (CC) is one of the leading causes of cancer death in sub-Saharan Africa (SSA), where CC burden is worsened by HIV epidemics.

- In 2020, the World Health Organization (WHO) announced its global strategy for the elimination of CC as a public health threat with targets for vaccination, screening, and treatment.

- Given that large cohorts of women remain unprotected by vaccines, we sought to monitor temporal trends in CC screening coverage, screening coverage by HIV status, rates of re-screening, and pre-cancer treatment coverage.

### What did the researchers do and find?

- We systematically analyzed and extracted data from 52 population-based surveys across 28 countries and used a multilevel Bayesian modeling framework to estimate CC screening coverage by age, HIV status, country, and region. We also examined the proportion of women screened twice by the age of 45 and cervical pre-cancer treatment coverage.

- Overall, only 1 in 7 women aged 30 to 49 years were estimated to have been ever screened for CC in 2020. Over 2000 to 2020, we found that CC screening coverage increased in Southern African but not much in Eastern and Western/Central Africa.

- Women living with HIV were more likely to be screened than women without HIV in all regions except Southern Africa.

- In 2020, 1 in 8 women were estimated to have been screened at least twice by the age of 45, much lower than the 70% target.

- Among the 4 countries with information on pre-cancer treatment coverage, 84% of women who received a positive CC test were estimated to have undergone pre-cancer treatment.

### What do these findings mean?

- Most women are not being reached by CC screening programs and none of the countries have reached WHO's targets for CC screening.

- Alongside expansion of HPV vaccination programs, action needs to be taken to improve and address barriers towards screening, including strengthening data collection systems, if CC is to be eliminated.

## Introduction

With approximately 604,000 new cases and 342,000 deaths reported in 2020, cervical cancer (CC) is the fourth most common cancer in women globally and the leading cause of cancer death in women in sub-Saharan Africa [1]. Low- and middle-income countries are disproportionately affected by the disease as they account for over 80% of the global CC burden, with sub-Saharan Africa having the highest age-standardized incidence and mortality rates in 2018 [2]. This high burden can be partially attributed to the existence of a syndemic (synergistic epidemic) between HIV and human papillomaviruses (HPV). HPV is the necessary cause of most CC [3] and the risk of developing CC is 6-fold higher in women living with HIV (WLHIV) in comparison to those without HIV [4].

Prevention and early treatment programs are highly effective measures that can reduce CC burden. Roll-out of national screening programs starting in the 1950s alongside effective HPV vaccinations beginning in the 2000s have resulted in dramatic reductions of disease incidence in high-income countries [5–8]. Countries in sub-Saharan Africa, however, face a range of challenges in implementing population-wide screening programs, including financial and logistical constraints [9]. This lack of access to critical prevention methods exacerbates CC burden. Globally, it is estimated that, without further interventions, annual CC deaths will rise to 443,000 in 2030 with 90% of the mortality occurring in sub-Saharan Africa [10].

In 2020, the World Health Organization (WHO) adopted a global strategy for the elimination of CC as a public health threat by 2030. This strategy included the "90-70-90" targets that calls for 90% of all girls to be vaccinated against HPV by 15 years of age, 70% of all women screened with a high-performance test once by age 35 and again by 45, and 90% of all pre-cancers treated and invasive cancer cases managed by 2030 [11]. Alongside this strategy, the WHO also released new screening and treatment recommendations. These recommendations indicated that screening should be prioritized for women 30 to 49 years among the general population and 25 to 49 years among WLHIV [12]. When high-performance tests are unavailable (i.e., HPV DNA tests), which is the case for many countries, screening is recommended every 3 years with visual inspection with acetic acid (VIA) or cytology [12]. Despite the importance of screening frequency in the WHO recommendations, little data exists on the proportion of women who were screened once and those screened at least twice.

Understanding trends in screening coverage and the subsequent screening care cascade (i.e., re-screening and treatment) is crucial to evaluate progress towards CC elimination. Despite this, estimates regarding screening and treatment, notably among WLHIV—an important priority group—are limited. Leveraging data from population-based surveys conducted in sub-Saharan Africa, our overall aim was to benchmark progress towards the screening and treatment goals by (1) examining overall, regional, and national trends in CC screening coverage (lifetime and past 3-year) by HIV status; (2) estimating the proportion of women screened twice before the age of 45; and (3) investigating CC treatment coverage among those with pre-cancerous lesions.

## Methods

### Data sources

We performed searches for nationally representative population-based surveys with data on CC screening or treatment coverage. Specifically, we searched data catalogs (i.e., the *Global Health Data Exchange*, the *WHO Multi-Country Studies Data Archive*, and the *WHO NCD Microdata Repository*) and conducted Google engine and literature searches. Building from previous HIV testing and CC screening reviews [13,14], we also systematically reviewed the following surveys: *Demographic and Health Surveys* (DHS), *Population-Based HIV Impact Assessment* (PHIA), *Study on Global AGEing and Adult Health* (SAGE), *STEPwise Approach to NCD Risk Factor Surveillance* (STEPS), *World Health Surveys* (WHS), *Kenya AIDS Indicator Survey* (KAIS), and the *South Africa National HIV Prevalence, Incidence, Behavior and Communication Survey* (SABSSM). All of the collected population-based surveys utilized complex sampling techniques and had high response rates. This study followed the *Guidelines for Accurate and Transparent Health Estimates Reporting* (GATHER) (Table A in S1 Appendix) [15].

### Data pre-processing

Data on CC screening (i.e., lifetime screening, screening in the past 3 years, and screening in the past year) and pre-cancer treatment coverage (i.e., treatment after positive test result for pre-cancer) were extracted from survey data, applying their survey weights. These data were all self-reported and the specific survey questions can be found in Table B (S1 Appendix). Survey outcomes were summarized by country, year, 5-year age groups, recall period, and HIV serostatus, if available. Treatment coverage was summarized by country. Due to potential clustering within primary sampling units, a design effect was calculated using the individual-level data to estimate the surveys' effective sample size. In the few instances where individual-level data was unavailable, we abstracted the most granular estimates available from survey reports and used the survey-adjusted confidence intervals to estimate the effective sample size. Finally, limited data was available on screening modalities and treatment approaches which prohibited such investigations.

A conceptual overview of the methods used for estimates of the trends in CC screening coverage is presented in Fig 1.

### Regional and national estimates of cervical cancer screening coverage time trends

**Statistical analyses for the estimation of trends in screening coverage.** CC screening coverage for lifetime screening and screening in the past 3 years for women 25 to 49 years was modeled using a flexible Bayesian multilevel binomial logistic regression model. To increase the number of included surveys, self-reports of lifetime and past 3-year screening were

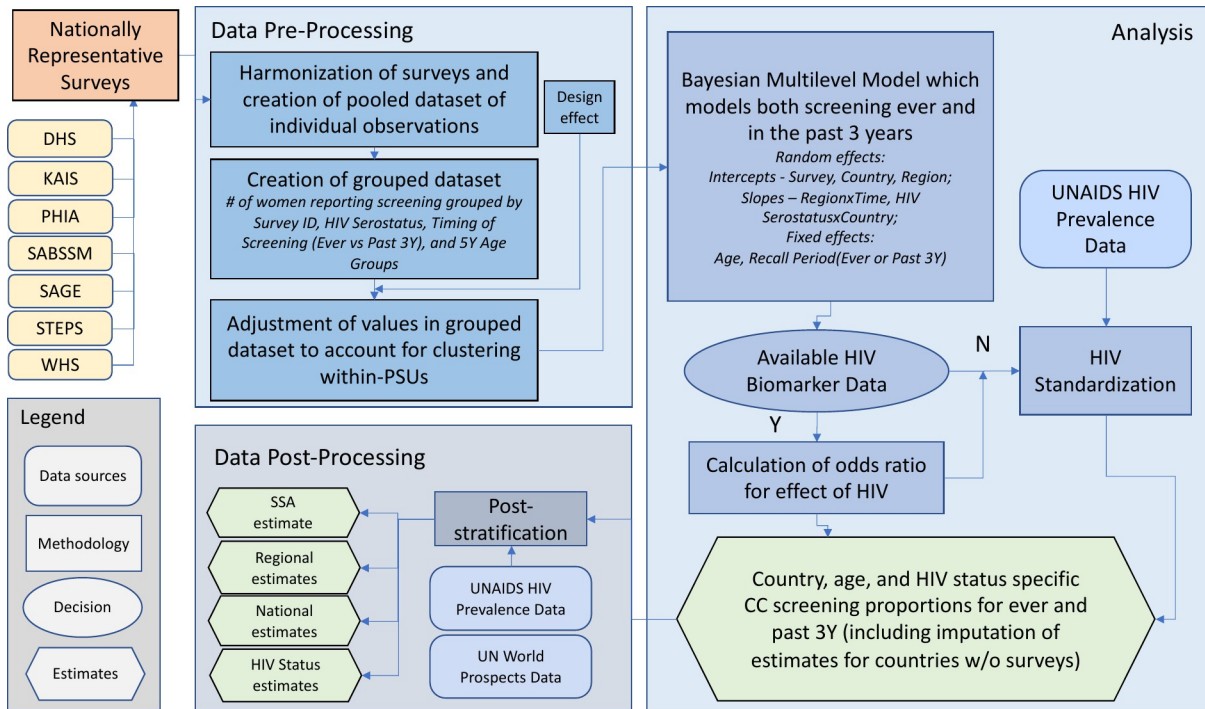

**Fig 1. Conceptual framework outlining data inputs, data pre-processing, statistical analyses, and data post-processing to estimate lifetime and past 3-year cervical cancer screening time trends by age, country, and HIV serostatus.** CC, cervical cancer; PSU, primary sampling units; SSA, sub-Saharan Africa; UNAIDS, The Joint United Nations Programme for HIV/AIDS; UN, United Nations. DHS, *Demographic and Health Surveys*; PHIA, *Population-Based HIV Impact Assessment*; SAGE, *Study on Global AGEing and Adult Health*; STEPS, *STEPwise Approach to NCD Risk Factor Surveillance*; WHS, *World Health Surveys*; KAIS, *Kenya AIDS Indicator Survey*; and the SABSSM, *South Africa National HIV Prevalence, Incidence, Behavior and Communication Survey*.

modeled jointly. The model structure, which was based on similar meta-regression models of health indicators [16,17] had 4 nested levels: survey, country, region, and sub-Saharan Africa. The model also accounted for age (by 5-year age groups), calendar year (continuous), recall period (lifetime versus past 3-year), and HIV serostatus (see Text A in S1 Appendix for model equations). Here, we defined regions based on the 2015 *Global Burden of Disease* classification (apart from Mauritius coded as an Eastern African country). Given the few surveys available in Western and Central Africa, these 2 regions were combined into 1 (i.e., Western/Central Africa).

Information regarding HIV serostatus was not collected in most surveys. To include surveys with and without information on HIV serostatus, we adopted a standardization approach [17]. To perform this, we included random slopes (nested levels: country, region, overall) to estimate odds ratios for screening coverage among WLHIV compared to women without HIV. For observations without information on HIV serostatus, we assumed the overall screening coverage was a weighted average of screening among WLHIV and those without HIV using UNAIDS HIV prevalence estimates [18] (Text A in S1 Appendix).

Model performance was assessed through posterior predictive checks and in-sample comparisons (Text A in S1 Appendix). Alternative model specifications, with different nested structures of random effects and addition of covariates such as existence of national screening programs and gross national incomes were also considered. Additionally, we examined the robustness of our results to restricting the study period to 2010 to 2020 and to the inclusion of a fixed effect for the WHS, the most common survey type informing trends in the early 2000s

(*Text A in S1 Appendix*). The "*rstan*" package [19] was used to fit models and the R statistical software [20] employed for all analyses.

## Post-processing of modeled estimates of screening coverage

Our fitted model provided lifetime and past 3-year screening coverage estimates by country, year, 5-year age groups, and HIV serostatus. To estimate screening coverage for broader age groups (i.e., the general population of women 30 to 49 years and WLHIV 25 to 49 years), combined HIV serostatus, and higher levels of aggregation (i.e., country, region, and sub-Saharan Africa), we pooled strata-specific estimates using post-stratification. Specifically, for each country and year, we took into consideration the underlying age distribution of the population using the *UN World Population Prospects* (2019 revision) [21] and its HIV prevalence using the UNAIDS HIV prevalence estimates [18]. In other words, when pooling strata (i.e., country, year, 5-year age group, and HIV serostatus), the population weight of each strata was considered. For countries without any surveys, screening coverages were imputed based on regional averages obtained from the model and we considered the additional uncertainty by sampling through the posterior distribution of country-level random effects (Text A in S1 Appendix). We provided estimates for each region and sub-Saharan Africa overall using all data available; however, country-level estimates were only presented for countries with at least 2 surveys.

## WHO recommendations for re-screening frequency

Available population-based surveys did not have direct re-screening information, thus it was not possible to empirically estimate the WHO goal of screening twice by age 45. To address this issue, we used life table methods [13,22] (Text B in S1 Appendix).

Our preliminary analyses suggested that the rate of first-time screening and the rate of re-screening were not the same. As such, we first estimated a rate ratio between the rate of re-screening (i.e., the rate at which women already screened once are screened again) and the rate of first-time screening for each region. (Due to limited data, Western, Central, and Eastern Africa were grouped into 1 region.) Estimating these rate ratios ideally requires longitudinal follow-up data. As this was unavailable, we compared increases in lifetime screening between successive 1-year age groups with the fraction of women reporting screening in the past year. We related the 2 values to obtain the rate ratios using a Bayesian multilevel model (see Text B in S1 Appendix for model equations). Simulations suggested that the rate ratio could be correctly estimated if there were no strong cohort or period effects. If screening rates changed through time, simulations indicated that biases could be minimized by restricting analyses to younger women (i.e., 18 to 29 years old; Text B and Figs H and I in S1 Appendix).

Using our estimated regional rate ratios (i.e., the ratio between rate of re-screening in a region over the rate of first-time screening in a region) alongside previously calculated age-, country-, and time-specific estimates of past 3-year screening coverage, we were then able to obtain age-, country-, and time-specific re-screening and first-time screening rates that were then included in our life tables (Text B in S1 Appendix). Specifically, we subjected a cohort of women aged 30 in 2005 to these calculated screening rates and estimated the proportion that would have been screened twice when they reached age 45 in 2020. To check the robustness of these estimates, lifetime screening estimates taken from the life tables were compared to those obtained from the time trends model described above. Additionally, sensitivity analysis using various rate ratio values on our life table methods was performed. (Text B in S1 Appendix).

### Estimating cervical pre-cancer treatment coverage

Survey-level pre-cancer treatment coverage was estimated among women reporting an abnormal screening result that was not suspected to be cancer. That is, women who have been screened and are suspected to have pre-cancerous lesions. Pre-cancer treatment coverage was meta-analyzed using a Bayesian logistic regression model with random effects for country (Text C in S1 Appendix).

### Ethics

In each survey, informed consent was obtained for all participants or their guardians. The specific consent procedures are described in the individual survey reports listed in Table B (S1 Appendix). Ethics approval for secondary data analyses was obtained from the Institutional Review Board of McGill University (A03-M19-20A).

## Results

### Survey characteristics

A total of 52 population-based surveys from 28 sub-Saharan African countries conducted between 2000 and 2020 were used for our inferences (Fig 2). Individual-level data was available for 49/52 of these surveys. Tabulations from reports were used for the remaining 3 surveys: Burkina Faso STEPS 2013, Mozambique STEPS 2015, and Zimbabwe PHIA 2020 (only WLHIV). Among these 52 surveys, only 4 had information regarding pre-cancer treatment coverage. Information on both lifetime and past 3-year screening was available from 15/52 surveys, 19/52 surveys had data on lifetime screening only, and 18/52 surveys on screening in the past 3 years only (Fig 2). A total of 14 countries had 2 or more surveys and 16/52 surveys had information on HIV serostatus. Additionally, 14/52 surveys had information regarding screening in the past year (used to estimate rate ratios for re-screening). In total, 151,338 women aged 25 to 49 years were included in the screening analysis and 113 women (without age restriction) were included in the CC treatment analysis.

### Regional and national estimates of cervical cancer screening coverage trends

Overall, our results suggested that lifetime CC screening among women aged 30 to 49 years remained constant over the 2000 to 2020 period in sub-Saharan Africa (Fig 3): starting from 14% (95% CrI: 8% to 25%) in 2000 to 14% (95% CrI: 11% to 21%) in 2020 (Table 1). Over these 2 decades, only Southern Africa was estimated to have increased its screening coverage. In Eastern and Western/Central Africa, coverages appeared to be minimally changing as trends were found to be slightly increasing and slightly decreasing, respectively, with large uncertainties (Fig 3).

In 2020, Southern Africa had the highest estimated CC screening coverage, with 51% (95% CrI: 40% to 62%) of women estimated to have been screened in their lifetime (Table 1). This was much lower in Eastern and Western/Central Africa regions with 13% (95% CrI: 9% to 19%) and 6% (95% CrI: 2% to 21%), respectively (Table 1). There were important country-level variations: Benin had the lowest screening coverage in 2020, with 1% (95% CrI: 0% to 2%) of women 30 to 49 ever screened, whereas South Africa had the highest coverage at 56% (95% CrI: 43% to 69%) (Fig 4, Table 1, and Fig B in S1 Appendix). Overall, most women who have ever been screened in 2020 (14%; 95% CrI: 11% to 21%) reported that their last screening occurred in the past 3 years (10%; 95% CrI: 8% to 15%) (Table 1 and Table C in S1 Appendix). Additionally, results from our sensitivity analyses yielded similar inferences (Figs C–E in

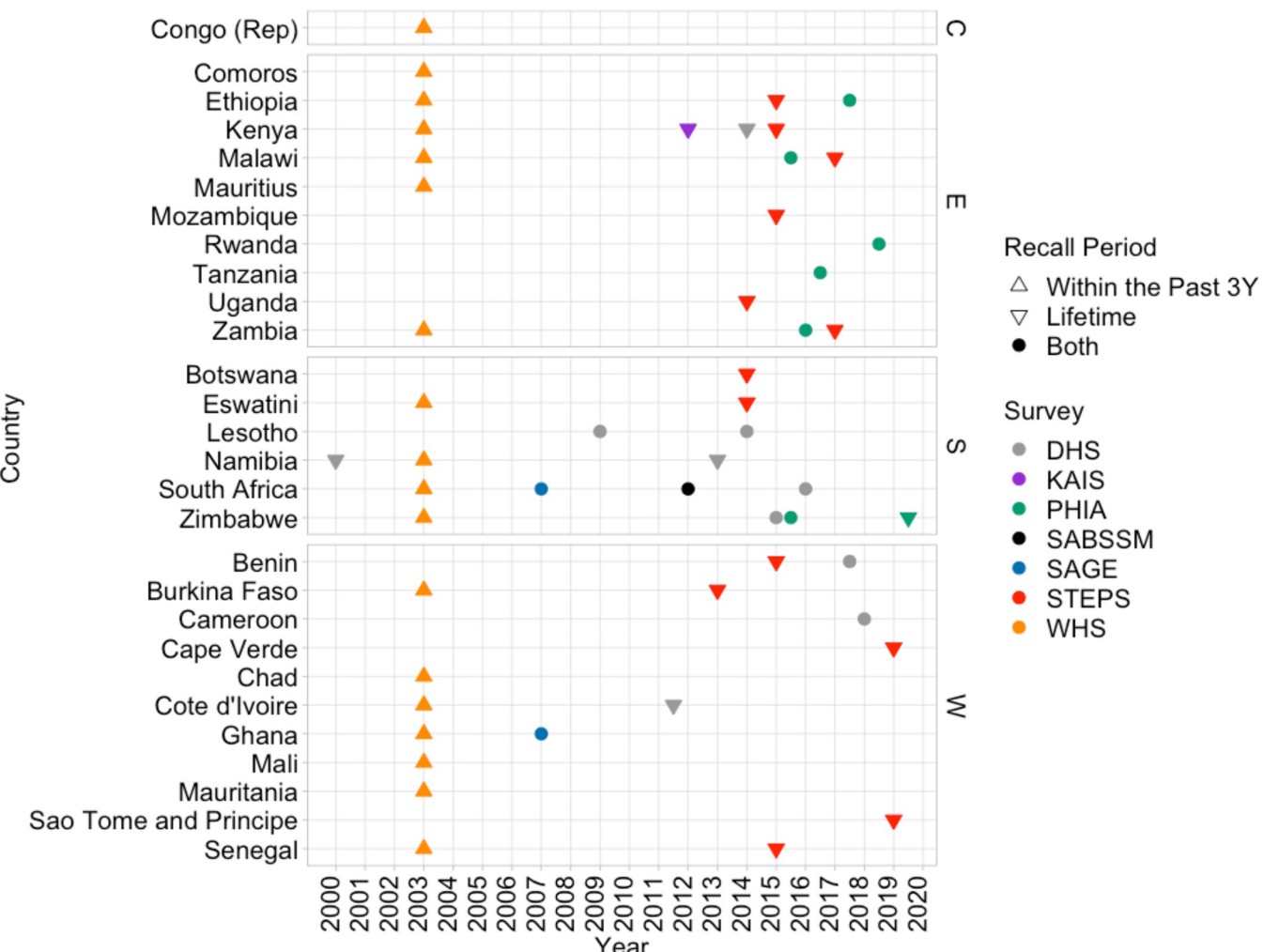

**Fig 2. Survey data availability plots.** Available surveys with information on cervical cancer screening by survey type, recall period, and region (C, Central Africa; E, Eastern Africa; S, Southern Africa; W, Western Africa). Colors represent survey type and shape represents recall period. (DHS, *Demographic and Health Surveys*; KAIS, *Kenya AIDS Indicator Survey*; PHIA, *Population-based HIV Impact Assessment*; SABSMM, *South Africa National HIV Prevalence, Incidence, Behavior and Communication Survey*; SAGE, *Study on Global AGEing and Adult Health*; STEP, *STEPwise Approach to NCD Risk Factor Surveillance*; WHS, *World Health Surveys*).

S1 Appendix). Finally, model validation through posterior predictive checks and in-sample comparisons suggested good model fit to the data (Fig F and Table D in S1 Appendix).

## Coverage of cervical cancer screening by HIV serostatus

WLHIV were found to have equal or greater odds of being screened (Table 2 and Fig B in S1 Appendix). In countries outside of Southern Africa, the adjusted odds ratio of women reporting screening among WLHIV varied between 1.7 in Kenya (95% CrI: 1.1 to 2.4) to 2.7 in the Tanzania (95% CrI: 2.2 to 3.3). In almost all countries in Southern Africa, except Zimbabwe (OR = 1.3; 95% CrI: 1.2 to 1.4), the ORs were close to unity.

Overall, 30% (95% CrI: 24% to 37%) of WLHIV aged 25 to 49 years had ever been screened for CC in 2020, compared to 11% (95% CrI: 8% to 18%) of women without HIV. This large difference is attributable to the higher prevalence of HIV in Southern Africa where screening

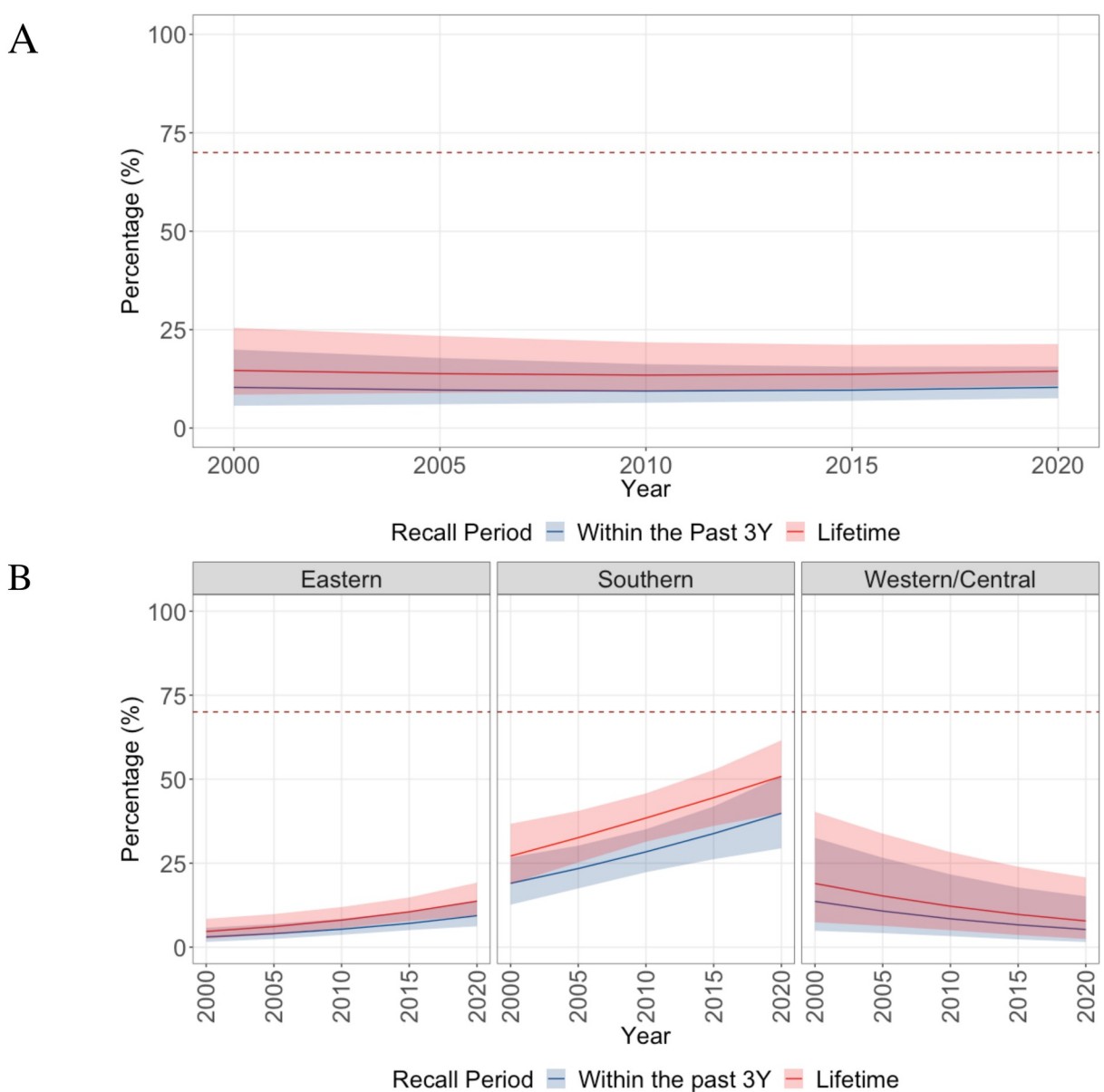

**Fig 3. Overall and regional-level trends in lifetime and past 3 years CC screening coverage among women aged 30–49 years between 2000–2020 in sub-Saharan Africa.** (A) Overall time trends for proportion of women screened. (B) Time trends for proportion of women screened by region. The red trendline represents the median lifetime screening trends. The blue trendline represents the median screening trends for screening in the past 3 years. The dotted red line represents the 70% screening goal set by the WHO. Shaded regions represent the 95% CrIs. CC, cervical cancer; CrI, credible interval; WHO, World Health Organization.

coverage is also higher. Among women 25 to 49 years in Eastern Africa, 19% (95% CrI: 14% to 25%) of WLHIV had ever been screened in 2020, compared to 11% (95% CrI: 8% to 17%) among women without HIV. In Western/Central Africa, 12% (95% CrI: 5% to 30%) of WLHIV compared to 6% (95% CrI: 2% to 19%) of women without HIV had ever been screened. In Southern Africa, screening coverage for both groups were similar with 49% (95% CrI: 35% to 61%) of WLHIV having ever been screened compared to 47% (95% CrI: 37% to 57%) of women without HIV (Table 2). Finally, most women who were screened in their lifetime were found to be screened in the past 3 years (Table 2).

**Table 1. Overall, regional, and national-level estimates for the proportion of women aged 30–49 years old screened for CC in their lifetime and in the past 3 years in 2020.** National-level estimates are presented only for countries with 2 or more surveys. (Results broken down by 5-year age groups can be found in Table C in S1 Appendix).

| Regions and countries | Proportion of women 30–49 years old screened for CC, median (95% CrI) | |
|---|---|---|
| | Lifetime | Past 3 years |
| **Overall**[*] | 14% (11%–21%) | 10% (8%–16%) |
| **Western/Central Africa**[*] | 6% (2%–21%) | 4% (2%–15%) |
| Benin | 1% (0%–2%) | 0% (0%–1%) |
| Burkina Faso | 4% (2%–11%) | 3% (1%–7%) |
| Côte d'Ivoire | 2% (1%–6%) | 1% (1%–4%) |
| Ghana | 3% (1%–7%) | 2% (1%–5%) |
| Senegal | 8% (4%–18%) | 5% (3%–12%) |
| **Eastern Africa**[*] | 13% (9%–19%) | 9% (6%–14%) |
| Ethiopia | 6% (3%–9%) | 4% (2%–6%) |
| Kenya | 21% (14%–30%) | 14% (9%–21%) |
| Malawi | 17% (11%–25%) | 12% (7%–18%) |
| Zambia | 23% (15%–32%) | 16% (10%–23%) |
| **Southern Africa**[*] | 51% (40%–62%) | 40% (29%–51%) |
| Eswatini | 25% (15%–40%) | 17% (10%–29%) |
| Lesotho | 20% (13%–32%) | 14% (8%–23%) |
| Namibia | 50% (34%–66%) | 39% (24%–55%) |
| South Africa | 56% (43%–68%) | 44% (32%–58%) |
| Zimbabwe | 30% (22%–40%) | 21% (15%–30%) |

[*]The overall and region-specific estimates consider the uncertainty and the population sizes of countries without surveys.

CC, cervical cancer; CrI, 95% credible intervals.

## Proportion of women aged 45 years of age screened at least twice

Using 14 surveys across 11 countries with information on both lifetime and past-year screening coverage, we estimated that the rate ratio of CC re-screening versus first-time CC screening was largely above one in all instances. That is, women who have already been screened once are screened again at a greater rate than women who have never been screened. For Western/Central/Eastern Africa, we estimated the rate ratio to be 34.1 (95% CrI: 16.8, 60) and for Southern Africa, we estimated the rate ratio to be 21.2 (95% CrI: 4.7, 64.9).

Among women aged 45 in 2020 in sub-Saharan Africa who have ever been screened, we estimated that 73% (95% CrI: 61% to 80%) of them have been screened at least twice. This corresponds to 12% (95% CrI: 9% to 18%) of women aged 45 having been screened twice in 2020 (Fig 5). Here again, there is significant between-country variation, and estimates are associated with large uncertainties. Validity checks suggest that pooled estimates for having been screened for CC at least twice are robust. However, the proportion of women screened twice in Southern Africa could be slightly underestimated (Text B and Fig J in S1 Appendix). Altogether, the results qualitatively suggest that women who have been screened once are likely to be screened a second time. Given the high proportion of women who have never been screened however, none of the countries were close to reaching the target of 70% of women screened for CC twice by the age of 45 in 2020.

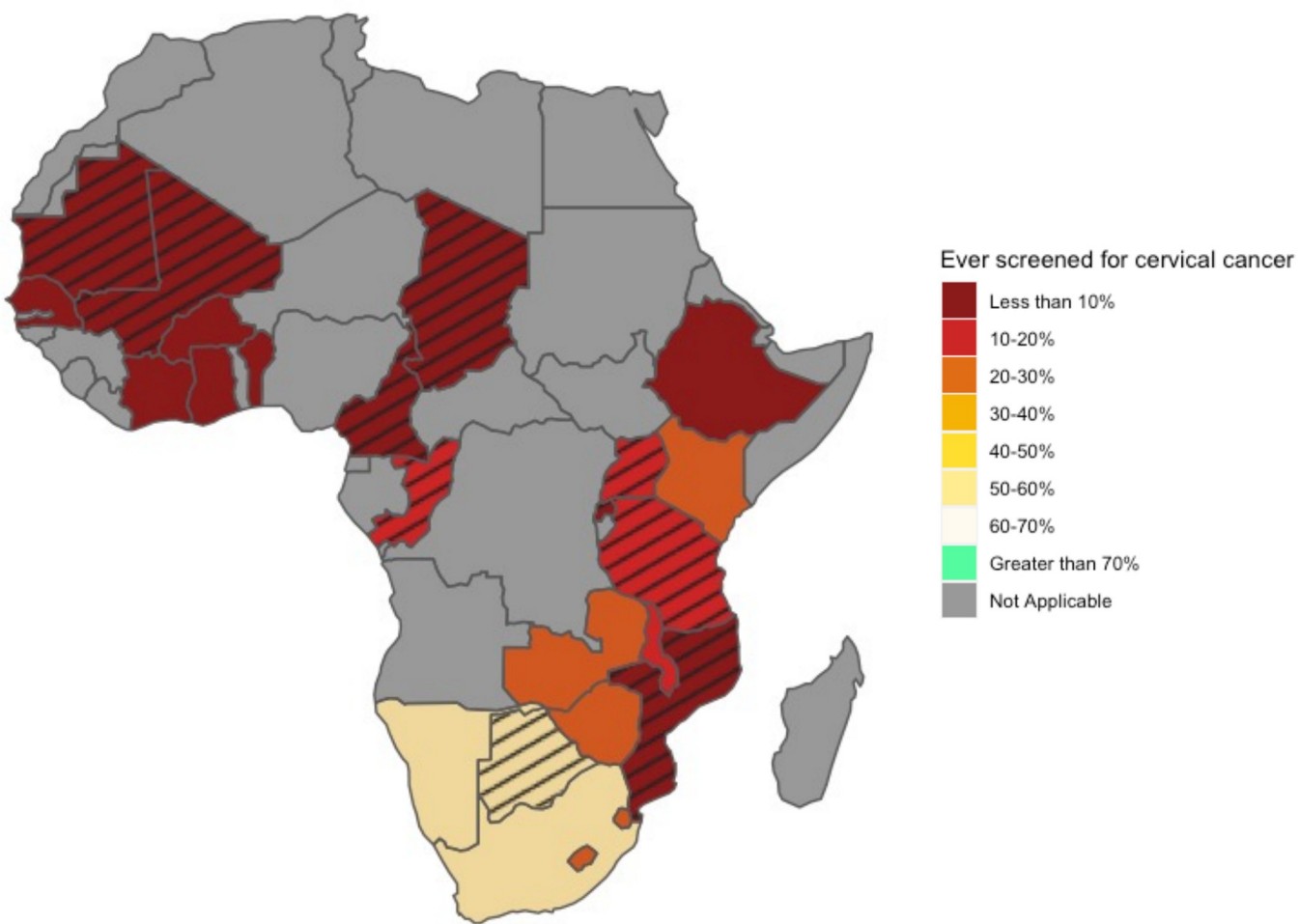

**Fig 4. Map of the percentage of women aged 30–49 years reporting having been screened in their lifetime for cervical cancer in 2020.** Gray-colored countries do not have any survey data or are located outside of sub-Saharan Africa, hatched areas represent countries for which estimates are extrapolated from 1 survey, and non-gray and non-hatched countries have data from 2 or more surveys. The base layer for this figure was obtained from the "world" dataset from the spData R package (https://cran.r-project.org/web/packages/spData/spData.pdf) that takes their data from Natural Earth. The terms of use for Natural Earth data can be obtained here: http://www.naturalearthdata.com/about/terms-of-use/.

## Coverage of cervical pre-cancer treatment among women with a positive screen result

Four countries had information on treatment coverage among women who had received an abnormal screening result that was not suspected to be cancer. Overall, the combined proportion of women who underwent cervical pre-cancer treatment across these 4 countries was 84% (95% CrI: 70% to 95%). Despite wide uncertainties, pre-cancer treatment coverage was lowest in Malawi (77%; 95% CrI: 60% to 88%), followed by Cape Verde (82%; 95% CrI: 55% to 94%), Tanzania (90%; 95% CrI: 66% to 98%), and Zambia (90%; 95% CrI: 65% to 98%) (Fig 6).

## Discussion

Using data from 52 population-based surveys from 28 countries in sub-Saharan Africa, we estimated that 14% of women aged 30 to 49 years had ever been screened for CC in 2020.

**Table 2. Country-level odds ratios of the impact of HIV serostatus on CC screening and regional and country-level estimates of lifetime and past 3-year CC screening among women aged 25–49 living with and without HIV in 2020.**

| Country | OR** (95% CrI) | Ever screened in 2020 (95% CrI) | | Screened in the past 3 years in 2020 (95% CrI) | |
|---|---|---|---|---|---|
| | | Women living with HIV | Women not living with HIV | Women living with HIV | Women not living with HIV |
| **Overall*** | 1.6 (0.3–12.8) | 30% (24%–37%) | 11% (8%–18%) | 23% (18%–29%) | 8% (6%–14%) |
| **Western/Central Africa*** | 2.1 (1.1–4.1) | 12% (5%–30%) | 6% (2%–19%) | 8% (3%–23%) | 4% (1%–14%) |
| Cameroon | 1.9 (1.2–3.2) | 10% (4%–20%) | 5% (2%–9%) | 7% (3%–14%) | 3% (1%–6%) |
| Côte d'Ivoire | 2.3 (1.1–5.3) | 5% (2%–12%) | 2% (1%–5%) | 3% (1%–8%) | 1% (0%–3%) |
| **Eastern Africa*** | 2.0 (1.6–2.6) | 19% (14%–25%) | 11% (8%–17%) | 14% (10%–19%) | 8% (5%–13%) |
| Ethiopia | 2.2 (1.7–3.1) | 10% (6%–17%) | 5% (3%–8%) | 7% (4%–11%) | 3% (2%–5%) |
| Kenya | 1.7 (1.1–2.4) | 27% (16%–39%) | 19% (12%–27%) | 19% (11%–30%) | 13% (8%–19%) |
| Malawi | 1.7 (1.4–2.0) | 21% (13%–31%) | 14% (9%–21%) | 15% (9%–23%) | 10% (6%–15%) |
| Rwanda | 2.1 (1.5–3.0) | 8% (4%–16%) | 4% (2%–8%) | 5% (2%–10%) | 2% (1%–5%) |
| Tanzania | 2.7 (2.2–3.4) | 19% (10%–33%) | 8% (4%–15%) | 13% (7%–24%) | 6% (3%–11%) |
| Zambia | 1.9 (1.6–2.2) | 29% (18%–41%) | 18% (11%–27%) | 21% (13%–31%) | 13% (8%–19%) |
| **Southern Africa*** | 1.1 (0.8–1.5) | 49% (35%–61%) | 47% (37%–57%) | 39% (27%–52%) | 37% (28%–48%) |
| Lesotho | 1.0 (0.8–1.3) | 19% (11%–32%) | 18% (11%–29%) | 13% (7%–23%) | 12% (7%–21%) |
| Namibia | 0.9 (0.7–1.2) | 46% (28%–63%) | 47% (31%–63%) | 36% (21%–54%) | 37% (23%–53%) |
| South Africa | 0.9 (0.8–1.0) | 53% (38%–67%) | 53% (41%–65%) | 42% (28%–57%) | 43% (31%–55%) |
| Zimbabwe | 1.3 (1.2–1.4) | 32% (22%–44%) | 26% (19%–35%) | 23% (15%–34%) | 19% (13%–27%) |

*For the proportion of women ever screened in 2020, the overall and region-specific estimates consider the uncertainty and the population sizes of countries without surveys.

**The ORs are adjusted for age, time, and the type of recall period for screening (lifetime versus past 3 years).

CC, cervical cancer; CrI, 95% credible intervals; OR, odds ratio.

Despite regional variations, overall screening coverage remained stagnant over the last 2 decades with only Southern Africa witnessing important increases. WLHIV were more likely to be screened in their lifetime compare to women without HIV in all regions, except in Southern Africa. Additionally, we found that women who had been screened once were more likely to be screened again, and we estimated that in 2020, 12% of women aged 45 had been screened at least twice: well below the WHO's 70% target. Finally, among the 4 countries with available data, 84% of women tested to have pre-cancerous lesions reported undergoing treatment. Although this value is approaching the WHO's 90% recommendation, low screening coverages suggest that many women with pre-cancerous lesions have likely not been screened and thus have not received treatment. Additionally, treatment data has not been collected in most countries and heterogeneity in pre-cancer treatment coverage in other regions likely exists.

To eliminate CC, it is essential for countries to develop adequate national primary (e.g., HPV vaccinations) and secondary (e.g., CC screening) prevention programs. National vaccination programs have been recently scaled-up among girls but two-thirds of countries in sub-Saharan Africa had yet to implement a program in 2020 [23,24]. Currently, it is estimated that 20% of girls in sub-Saharan Africa have been fully vaccinated against HPV [23]. Despite the importance of primary prevention, vaccinations are only prophylactic and will not cure women already infected with HPV. As such, quality screening programs that are linked to treatment remain imperative to prevent CC in the decades to come for large cohorts of women who have not been vaccinated prior to sexual debut [25]. Our findings suggest that screening coverage has remained stagnant over the last 2 decades in most regions of sub-Saharan Africa. Efforts are needed to strengthen and rapidly scale-up screening programs if the 2030 WHO targets are to be met.

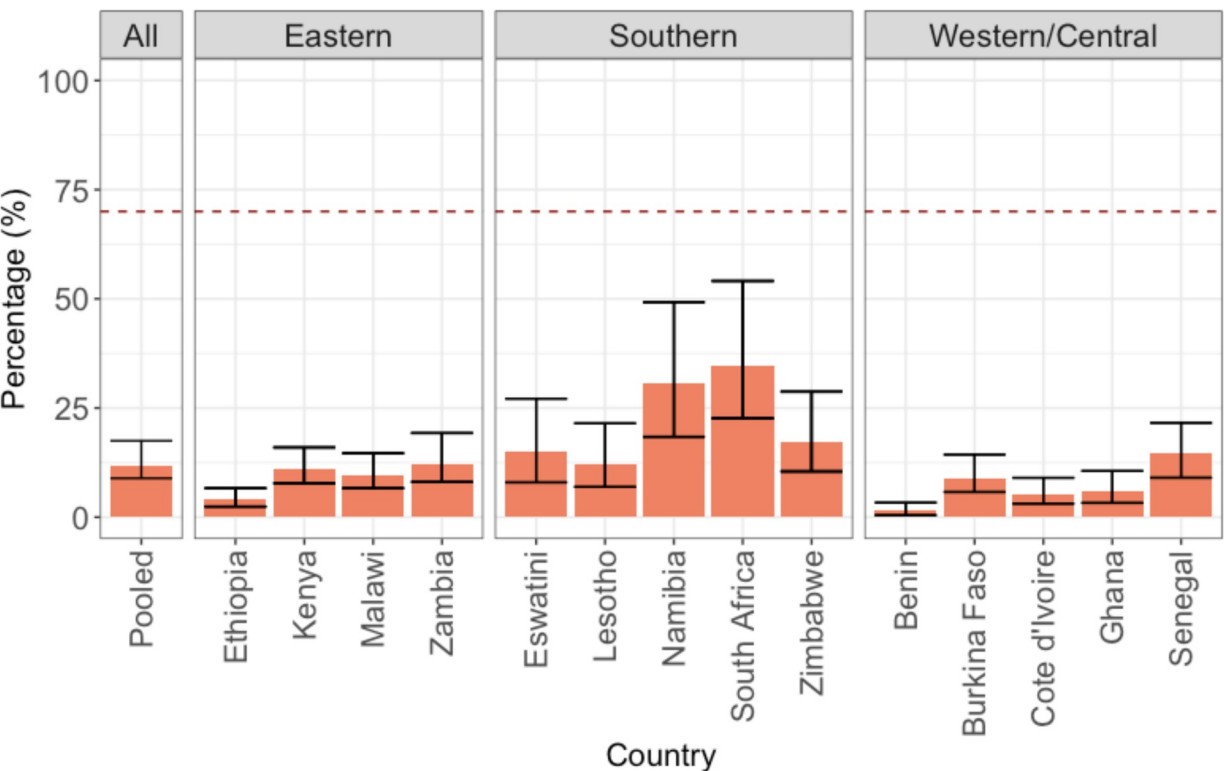

**Fig 5. Estimates of the percentage of women aged 45 years in 2020 who have been screened twice for CC in 2020.** The pooled estimate is for the whole sub-Saharan region. The dotted red line represents the 70% screening goal set by the WHO. The height of the bars represents the median estimates and the error bars the 95% CrIs. CC, cervical cancer; CrI, credible interval; WHO, World Health Organization.

Many barriers currently exist to achieving higher screening coverages in several sub-Saharan African countries. These include, but are not limited to, various financial, social, and structural constraints such as lack of CC knowledge, competing interests, and absence of political will to invest in CC screening programs, among others [9]. Greater financial resources, developed healthcare infrastructure, and historical attention to cervical cancer screening priorities may have contributed to higher screening coverage in Southern African countries. Additionally, integration of CC screening with existing health services has been a strategy utilized to address some of the barriers towards screening, notably integration with HIV care clinics allowing greater opportunity for WLHIV to be screened [9,26]. With renewed attention to the additional risks of CC development in WLHIV, recognition of WLHIV as a priority group for CC prevention, and recommendations for increased frequency of CC screening among WLHIV compared to those without HIV, initiatives such as HIV/CC service integration has been implemented in many countries in Africa, supported in part by The US President's Emergency Plan for AIDS Relief (PEPFAR) [26]. Although the impact of service integration is still being evaluated, this approach could partially explain why we found WLHIV in some regions had higher screening coverages compared to women without HIV.

It is important to note that the Coronavirus Disease 2019 (COVID-19) pandemic has likely exacerbated many barriers to CC screening. Disruptions to health services, changes in health-seeking behaviors [27,28] as well as indirect economic effects [29] have likely resulted in reduced access and demand for CC screening and treatment services in the short and long term. The COVID-19 pandemic has likely also widened inequities in screening and treatment

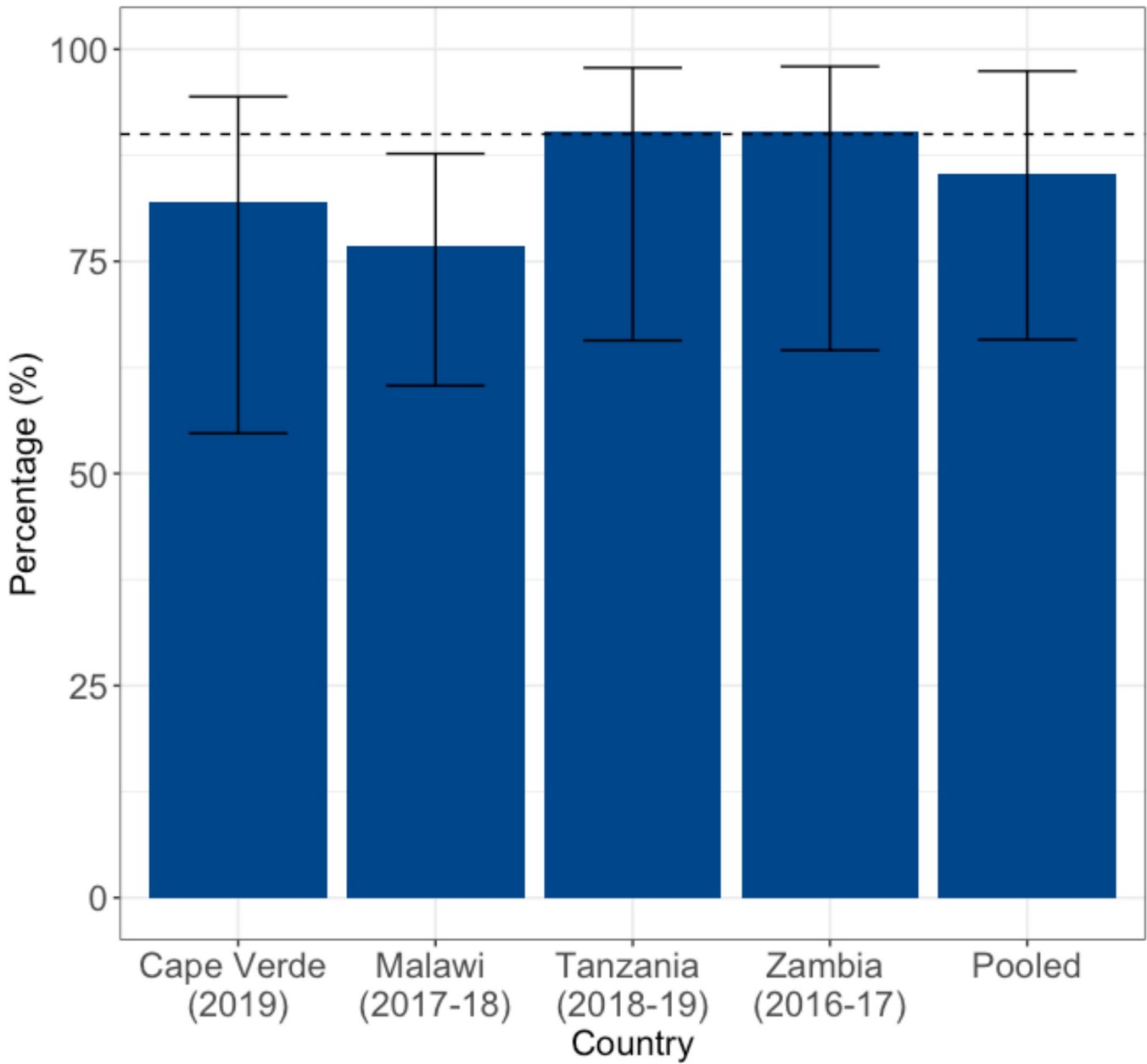

**Fig 6. Reported pre-cancer treatment coverage among women with an abnormal or positive screening result.** Country-level estimates are taken directly from the survey data. Pooled estimates are obtained from the model. The dotted blue line represents the 90% treatment goal set by the WHO. The height of the bars represents the median estimates and the error bars represent the 95% CrIs. CrI, credible interval; WHO, World Health Organization.

coverage. Empirical studies have found that disparities in screening coverage exist along various social determinants of health such as education and geography [14]. Our findings similarly suggest that only a small subset of the population may be screened and re-screened as we found that re-screening rates were many folds that of first-time screening. Although research regarding re-screening is limited, our findings are aligned with a 1993 study conducted in South Africa which found that certain rural workers were being over-screened while others were excluded altogether [30].

In general, more data should be collected on the full screening care cascade (i.e., re-screening and treatment). Information on re-screening in the literature is sparse and data on screening frequency is not collected in most population-based surveys. Our study estimated that overall, 73% of women who have been screened once have been screened a second time; however, important country-level variations likely exist. One study from Mozambique [31] found that only 29% of women 30 to 55 years had ever been screened more than once. On the other hand, a cohort study [32] in Harare, Zimbabwe found that over 70% of WLHIV were re-screened. Given the limitations of our methods to estimate re-screening rate ratios, information on screening frequency would enable more robust and granular estimates of re-screening patterns. Similarly, only 4 countries had surveys with information related to treatment. Linkage to treatment is imperative for effective screening and more data regarding treatment including treatment modalities and approaches used (e.g., use of screen and treat programs) must also be collected. Altogether, to monitor progress towards the CC elimination goals, survey instruments need to capture the entire CC screening care cascade.

Our results need to be interpreted considering this study's limitations. First, survey data on CC screening and treatment was relatively sparse. Only 14 countries had surveys from multiple years, and only 12 countries had information regarding HIV serostatus. Furthermore, the treatment analysis was based on the self-reports of only 113 women. This could also possibly lead to an overestimation of screening coverage as countries without data may have lower screening coverages and treatment coverages. We addressed this by using a multilevel statistical approach that allowed us to borrow strength across countries and to propagate the uncertainty to the model results. Similarly, the limited data regarding screening modalities and treatment approaches prevent their more detailed analysis. Secondly, our estimates of re-screening ratios are modeled from cross-sectional data, assuming no cohort or period effects. This assumption was likely violated for Southern Africa, despite limiting bias by focusing on the younger age groups. Nevertheless, the strong patterns observed suggest high rates of re-screening. Thirdly, different survey questionnaires asked slightly different questions. For example, some surveys asked specifically about Pap smears, while other asked about CC screening in general. Although there is limited information regarding primary screening modalities prior to 2015, to the best of our knowledge, when information was available, the primary method of screening within a country matched the screening method asked about in the survey [33]. Finally, our analysis relied on self-reported measures of screening and treatment coverage. Due to biases such as recall and social desirability bias, this may potentially lead to overestimation of our estimates [34]. A summary of the main assumptions and their justifications can be found in Table F (S1 Appendix).

Strengths of this investigation include the incorporation of population-based survey data from the greatest number of countries and sources in sub-Saharan Africa to date. Second, model validation suggested that we accurately reproduced empirical observations with appropriate propagation of uncertainty to model results. Third, by quantifying re-screening patterns, we provided valuable information on indicators to benchmark WHO elimination targets for CC screening that are currently limited. Finally, this study unequivocally established that screening coverage in 2020 is low and that there are important disparities between countries and regions.

In conclusion, screening and pre-cancer treatment coverages for CC are currently below the WHO elimination goals. To reach these goals by 2030 and eliminate this highly preventable disease, screening and treatment programs need to be scaled-up alongside HPV vaccination programs. Use of acceptable alternative modalities such as HPV self-testing [35] and effective roll-out strategies that educate and engage communities will be beneficial to improving screening and treatment coverage. Finally, to be able to effectively track progress towards these goals,

future studies and surveys should prioritize greater data collection along the CC screening and treatment cascade.

## Supporting information

**S1 Appendix. Additional information on surveys, model equations, supplemental model descriptions, sensitivity analyses, and model validation.**
(PDF)

## Acknowledgments

We are grateful to all participants of these population-based surveys for generously donating their time to complete the survey of which this analysis would not be possible without and for the surveying teams for making their data available.

## Author Contributions

**Conceptualization:** Lily Yang, Mathieu Maheu-Giroux.

**Data curation:** Lily Yang.

**Formal analysis:** Lily Yang, Mathieu Maheu-Giroux.

**Investigation:** Lily Yang, Caroline Hodgins, Mathieu Maheu-Giroux.

**Methodology:** Lily Yang, Marie-Claude Boily, Minttu M. Rönn, Dorcas Obiri-Yeboah, Imran Morhason-Bello, Nicolas Meda, Olga Lompo, Philippe Mayaud, Michael Pickles, Marc Brisson, Sinead Delany-Moretlwe, Mathieu Maheu-Giroux.

**Supervision:** Mathieu Maheu-Giroux.

**Validation:** Lily Yang, Mathieu Maheu-Giroux.

**Visualization:** Lily Yang, Marie-Claude Boily, Minttu M. Rönn, Dorcas Obiri-Yeboah, Imran Morhason-Bello, Nicolas Meda, Olga Lompo, Philippe Mayaud, Michael Pickles, Marc Brisson, Caroline Hodgins, Sinead Delany-Moretlwe, Mathieu Maheu-Giroux.

**Writing – original draft:** Lily Yang.

**Writing – review & editing:** Marie-Claude Boily, Minttu M. Rönn, Dorcas Obiri-Yeboah, Imran Morhason-Bello, Nicolas Meda, Olga Lompo, Philippe Mayaud, Michael Pickles, Marc Brisson, Caroline Hodgins, Sinead Delany-Moretlwe, Mathieu Maheu-Giroux.

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
