## [Editor Report · Decision Letter 0]

4 Aug 2022

Dear Dr Maheu-Giroux, 

Thank you for submitting your manuscript entitled "Regional and country-level trends in cervical cancer screening coverage in high HIV prevalence settings: a systematic analysis of population-based surveys in sub-Saharan Africa (2000-2020)" for consideration by PLOS Medicine.

Your manuscript has now been evaluated by the PLOS Medicine editorial staff and I am writing to let you know that we would like to send your submission out for external peer review.

Please re-submit your manuscript within two working days, i.e. by Aug 08 2022 11:59PM.

Kind regards,

Callam Davidson

Associate Editor

PLOS Medicine

---

## [Decision Letter · Decision Letter 1]

7 Sep 2022

Dear Dr. Maheu-Giroux,

Thank you very much for submitting your manuscript "Regional and country-level trends in cervical cancer screening coverage in high HIV prevalence settings: a systematic analysis of population-based surveys in sub-Saharan Africa (2000-2020)" (PMEDICINE-D-22-02613R1) for consideration at PLOS Medicine. 

Your paper was evaluated by an associate editor and discussed among all the editors here. It was also discussed with an academic editor with relevant expertise, and sent to independent reviewers, including a statistical reviewer. The reviews are appended at the bottom of this email and any accompanying reviewer attachments can be seen via the link below:

[LINK]

In light of these reviews, I am afraid that we will not be able to accept the manuscript for publication in the journal in its current form, but we would like to consider a revised version that addresses the reviewers' and editors' comments. Obviously we cannot make any decision about publication until we have seen the revised manuscript and your response, and we plan to seek re-review by one or more of the reviewers. 

We hope to receive your revised manuscript by Sep 26 2022 11:59PM. Please email us (plosmedicine@plos.org) if you have any questions or concerns.

We look forward to receiving your revised manuscript. 

Sincerely,

Callam Davidson, 

PLOS Medicine

plosmedicine.org

Please include continuous line numbering throughout your manuscript to facilitate review.

Please update your title to ‘Regional and country-level trends in cervical cancer screening coverage in sub-Saharan Africa (2000-2020): a systematic analysis of population-based surveys’.

In your Data Availability Statement, please confirm whether the data are freely available on request from the sources provided, or whether restrictions are in place. See https://journals.plos.org/plosmedicine/s/data-availability#loc-faqs-for-data-policy for more details.

Please include the study design (a systematic analysis of population-based surveys) In your Abstract Methods and Findings.

Thank you for providing the GATHER checklist. Please update Table S6 to use section name/paragraph number rather than page number.

Some of the text in Figure 1 is very small – please enlarge.

Please remove the ‘Role of the funding source’ section.

I could not find details of the specific consent procedures in Table S1. 

Please define the meaning of the shaded area in the Figure 3 legend.

Please confirm that the appropriate usage rights apply to the use of the map in Figure 4. Please see our guidelines for map images: https://journals.plos.org/plosmedicine/s/figures#loc-maps

Please define the meaning of the error bars in the Figure 5 and Figure 6 legends.

Please remove the Data Sharing, Declaration of interests, and Funding sections and instead ensure that all relevant information is captured in your responses to the respective sections of the submission form. 

Please use ‘et al.’ only after listing the first six authors in your references.

Please include ‘[preprint]’ in reference 15. See https://journals.plos.org/plosmedicine/s/submission-guidelines#loc-references for more details.

Comments from the reviewers:

Reviewer #1: GENERAL COMMMENTS

Yang et al pooled data from published surveys on cervical screening coverage in Sub-Saharan Africa, a subcontinent with high prevalence of HIV and the highest burden of cervical cancer in the world. Whereas very effective tools for screening (in particular several validated HPV assays) and management of screen+ women are available, access to screening and availability of affordable screening assays is problematic in this region.

WHO recommends 2 screenings with a high precision test (say HPV test) and re ach 70% coverage among women in the age 35-49Y and manage 90% of screen+ women and women with (pre-) cancer. Using the best available data completed with appropriate modelling, authors succeeded in estimating the coverage for ever screened, coverage for at least two screenings and treatment coverage.

A strong and innovative point of the paper is the attention for repeated screening and the treatment coverage.

A weak point is the lack of clarity of "rescreening rate ratios". The discussion can be shortened but otherwise some other important discussion points could be added as suggested below.

SPECICIF COMMENTS

ABSTRACT

Background: term ".. monitored re-screening by age 45" is not clear. % screened once and % screened (at least) twice looks easier to understand.

"High HIV prevalence setting" is not defined in the abstract.

INTRODUCTION

Together with ref 2 (dedicated to all cancers), the authors could cite the paper, dedicated to cervical cancer only: https://www.ncbi.nlm.nih.gov/pubmed/31812369

First sentence of 2nd §. A more international ref could be given to illustrate the trends of CC and how it changed through screening for instance (https://www.ncbi.nlm.nih.gov/pubmed/19695864 ). 

One of the refs that illustrates the impact of HPV vaccination on incidence of CC in young women: https://www.ncbi.nlm.nih.gov/pubmed/34741816

Last sentence of §3. Despite the importance of screening frequency in the WHO recommendations, little data exists on re-screening frequency in sub-Saharan Africa.

Also on the % screened once there is little data.

Suggestion:" llittle data exists on the proportion of women who were screened once or two times."

The introduction does not clarify "High HIV prevalence setting". This should be explained in the Introduction, or maybe the title should be changed. By reading the title I had the impression that settings (HIV clinics, sexual health centres, …) were the target of the paper. After reading the Intro, I understood that the whole female population of sub Saharan African countries (where HIV prevalence is high) is the target. 

MATERIAL & METHODS

Data processing

Page 4, line 2: treatment outcome: this usually means the % where treatment was successful or failed. Better wording would be: percentage treated or treatment coverage.

Regional and national estimates of cervical cancer screening

coverage time trends

Did the authors did not have one dbase where observed 2nd screening data were available to validate the modeled estimates?

Given the large number of figures, I suggest to move Fig 2 to the supplementary materials.

The colors used could be more contrasting to increase read-friendliness.

Authors present pooled estimates for sub-continents and for the whole SSA. Was a weighting used (proportional to the population size of the countries providing data) or was the pooling done using study size weights. A population weighting could be considered if not done. 

The authors should provide definition of "rate ratios" clarifying numerator and denominator, used to compute the proportion women ever screened who had at least 2 screenings. 

RESULTS

Figure 3. Please add information on the credibility intervals.

Figure 4: the contrast between color schemes is weak. Please use a stronger contrast scheme. 

Coverage of cervical cancer screening by HIV serostatus

Page 14, last line of §1: "ORs were close to the null". Should be changed to ".. close to unity".

Proportion of women aged 45 years of age screened at least twice

Given the lack of definition of "rate ratio" in the Methods it is difficult to understand "…we estimated that the rate ratio of re-screening versus first-time screening was largely

above one in all instances.". I intuitively expected a value <1, meaning that not all ever screened women would have had a 2nd one. However, after reading the Results it becomes that my first intuitive understanding was wrong. 

Title of Figure 5. "Estimates of women aged 45 years in 2020…". Change into: "Estimates of the percentage of women aged 45 years in 2020… among all women in the target age range". Credibility intervals are shown but not explained in the figure title.

DISCUSSION

Page 17, §2

"vaccinations are not prophylactic", change into vaccinations are only prophylactic.

I am missing the following three discussion points:

- probably the screening coverage and treatment coverage estimates are better in countries with data compared to those without data; so the pooled estimates might be worse for the whole SSA

-HPV testing on self-samples offers opportunities to increase coverage as concluded from a recent meta-analysis (https://www.ncbi.nlm.nih.gov/pubmed/30518635) including two trials conducted in Africa demonstrating its effectiveness.

- surveys collecting information regarding self-reported screening experience generally overestimate over-estimate screening coverage. 

Reviewer #2: Yang et al present the results of a series of analyses, bringing together data from a number of large-scale surveys conducted in sub-Saharan Africa, looking at cervical cancer screening and treatment over the first twenty years of this century. This review is interested in the use of statistics in the paper.

There is a huge amount of work here. The guiding aim of the work is to assess the current status and progress towards the WHO targets in terms of screening and treatment, but there is also an interest in the association with HIV status. The authors were able to access individual level data from most surveys, and use Bayesian models to estimate the parameters of interest. Where they do not have access to data to estimate what they need directly (re-screening by age 45), they employ an indirect method. Throughout, they endeavour to present their estimates with appropriate levels of uncertainty. Overall, I believe their statistical methods to be sound, the results quite well presented, and the results not overstated.

My comments are minor.

The abstract states that an estimate 14% of women aged 30-49 had ever been screened in 2020. The credible interval is given as 11-22%. However, in the results, on page 11 and in Table 1, the CI is given as 11-21%. I suggest a thorough QC.

The abstract describes rescreening rates as high, and then states that an estimated 12% of women had been screened twice by age 45 in 2020. Whilst I can see what is meant, this is confusing at first, since 12% is not "high".

Looking at Figure 2, it is notable that almost all the surveys prior to 2010 were from the WHS in 2003, with only four other surveys in that decade. Could this concentration of information around 2003 have an undue influence on the estimated time trends in the data? Were any sensitivity analyses carried out, perhaps looking only at data from 2010-2020? Perhaps a series of sensitivity analyses could be done, excluding each survey type in turn?

The data sharing statement includes a link to a clean dataset and program, but the link did not work for me. This is not critical to my review - I was looking out of interest - but it will need to be fixed.

Reviewer #3: Thank you for the opportunity to review this manuscript. Yang and colleagues used data from 52 population-based surveys from 28 sub-Saharan African countries examine temporal trends (2000-2020) in cervical cancer (CC) screening coverage. Data on the coverage of CC screening and treatment of pre-cancerous cervical lesions among women in sub-Saharan Africa are scarce and, therefore, this well-written paper addresses an important gap in the literature. The authors try to overcome the data limitations by applying Bayesian multilevel models, borrowing information from surveys in other countries. This is a valuable approach although, as the authors acknowledge, ultimately more primary studies examining CC screening and pre-cancer treatment coverage in sub-Saharan Africa are needed. The merit of the paper could be increased if some of the analyses were further stratified by HIV status, given that the authors refer to "high HIV prevalence settings" in the title. 

Please find specific comments on the individual sections below.

General:

1. The authors often refer to "treatment" or "CC treatment" instead of "cervical pre-cancer treatment." To avoid confusion, the authors should make sure to refer to pre-cancer treatment throughout all sections of the manuscript including the supplemental material.

2. It would be helpful if all supplemental materials were referenced in the manuscript.

Abstract:

1. Under "Methods and Findings" it would be helpful if the authors made clear that these were cross-sectional and not longitudinal surveys.

2. It would be good to mention the much higher screening coverage in Southern Africa compared to the other African regions.

3. The sentence "Rescreening rates were high, and it was estimated that 12% (95%CrI: 10-18%) of women had been screened twice or more by age 45 in 2020" is a bit confusing, as 12% does not seem high. I suggest rephrasing to clarify that the high rescreening rates refer to previously screened women, whereas the 12% refer to all women.

Introduction:

1. The word "between" in the last sentence of the introduction section can be deleted.

Methods: 

1. Data on "screening in the past year" extracted from the surveys but not really used in this analysis? Or was this information used in the life table method?

2. "Treatment outcomes" is not an ideal term for "treatment after positive test result for pre-cancer" as many readers will think of outcomes after pre-cancer treatment such as disease recurrence, adverse events, etc.

3. The limited data on screening modalities and cervical pre-cancer treatment approaches should also be acknowledged in the limitations section of the discussion (not only in the methods).

4. The authors state that country-level estimates were only presented for countries with at least two surveys. Did the two surveys have to assess the same CC screening recall period (lifetime vs past 3 years) or not?

5. The sentence "Our preliminary analyses suggested that first-time screening and re-screening were independent" needs further explanations. It seems counterintuitive that first-time screening and re-screening are independent.

6. Pre-cancer treatment information was also self-reported, correct? 

Results:

1. Would it be possible to show country-level CC screening trends stratified by both HIV status and recall period (lifetime vs past 3 years)? Typically, shorter screening intervals are recommended for women living with HIV. Therefore, it would be interesting to see how HIV status impacts CC coverage estimates by recall period. Or if not the trends, then maybe the CC coverage estimates reported in Table 1 could be stratified by both recall period and HIV status?

2. The authors found that in Southern Africa the odds of screening were similar among women with and without HIV. Does this only refer to lifetime screening or also screening in the past 3 years?

3. If the OR in Table 2 refers to "ever screened" (i.e., lifetime screening?), does it make sense to adjust for recall periods?

4. Would it be possible to stratify the estimates in Figure 5 by HIV status? As more frequent screening is recommended for women living with HIV, it would be interesting to see how HIV status impacts the re-screening rates.

5. It would be interesting to know a bit more about how the pre-cancer treatment coverage was assessed in the 4 available surveys. Did they ask about pre-cancer treatment within a certain time frame or at any point after a positive CC screening result?

6. The sentence "As this analysis was limited to women who had been screened and had received their screening result, the true number of women with pre-cancerous lesions who are treated is likely to be lower" rather belongs into the discussion section.

Discussion:

1. "Despite the importance of primary prevention, vaccinations are not prophylactic and will not cure women already infected with HPV." I assume the authors mean "not therapeutic" rather than "not prophylactic" (or want to delete the "not"). 

2. The authors mention that service integration could partially explain why they found higher CC screening coverage among women living with HIV in certain regions. The authors may also want to mention that earlier and more frequent screening is recommended for women living with HIV which may affect coverage estimates particularly among the younger age group. 

3. Could the authors elaborate more on potential reasons for the higher CC screening coverage in Southern Africa compared to the other regions?

4. The analysis is based on self-reported information. How do the authors expect self-reporting to affect their estimates? Should this be included in the limitations section?

5. The authors should highlight in the limitations section that their assessment of pre-cancer treatment coverage is based on 113 women only. 

Data sharing:

1. It is great that the authors make the code available on GitHub. Unfortunately, the provided link to the GitHub repository does not work. Is it really publicly accessible?

[LINK]

---

## [Decision Letter · Decision Letter 2]

4 Nov 2022

Dear Dr. Maheu-Giroux,

Thank you very much for re-submitting your manuscript "Regional and country-level trends in cervical cancer screening coverage in sub-Saharan Africa: a systematic analysis of population-based surveys (2000-2020)" (PMEDICINE-D-22-02613R2) for review by PLOS Medicine.

I have discussed the paper with my colleagues and the academic editor and it was also seen again by three reviewers. I am pleased to say that provided the remaining editorial and production issues are dealt with we are planning to accept the paper for publication in the journal.

[LINK]

We look forward to receiving the revised manuscript by Nov 11 2022 11:59PM.   

Sincerely,

Callam Davidson, 

Associate Editor 

PLOS Medicine

plosmedicine.org

Comments from Reviewers:

Reviewer #1: No further comments

Reviewer #2: Alex McConnachie, Statistical Review

I thank the authors for their responses to my earlier comments, and I am happy with the changes they have made.

One last thing. Lines 191-2 say that "This hierarchical structure allowed us to borrow statistical strength across observations". This is a very woolly phrase, and does not actually mean anything concrete. I would remove. There is a similar statement in the discussion, where it is better placed, rather than in the description of the statistical methods.

Reviewer #3: Please find two minor comments below.

Author summary:

1. Typo in "WLHIV were more likely to be screened than women without HIV in all regions expect Southern Africa." The word "expect" should be replaced by "except". 

Results:

1. The authors added a statement that WLHIV were recognized as a priority group for CC prevention. However, this does not explicitly mention that more frequent CC screening is recommended for WLHIV than for women without HIV. I think this is an important information given that the authors assess the impact of HIV serostatus on CC screening.

[LINK]

---

## [Editor Report · Decision Letter 3]

18 Nov 2022

Dear Dr Maheu-Giroux, 

On behalf of my colleagues and the Academic Editor, Professor Sanjay Basu, I am pleased to inform you that we have agreed to publish your manuscript "Regional and country-level trends in cervical cancer screening coverage in sub-Saharan Africa: a systematic analysis of population-based surveys (2000-2020)" (PMEDICINE-D-22-02613R3) in PLOS Medicine.

Please also remove the 'Contributors' section from the end of the main text - this information will be captured and published as metadata as part of the submission process in Editorial Manager. 

PRESS

Sincerely, 

Callam Davidson 

Associate Editor 

PLOS Medicine